# Stability criteria for the consumption and exchange of essential resources

Theo Gibbs [1]*, Yifan Zhang [2], Zachary R. Miller [3], James P. O'Dwyer [2]

**1** Lewis-Sigler Institute for Integrative Genomics, Princeton University, Princeton, New Jersey, United States of America, **2** Department of Plant Biology, University of Illinois, Urbana, Illinois, United States of America, **3** Department of Ecology & Evolution, University of Chicago, Chicago, Illinois, United States of America

\* tgibbs@princeton.edu

**Data Availability Statement:** All relevant data are within the manuscript and the Supporting information files. The code for generating the data is available on GitHub at https://github.com/theogibbs/essential-stability-criteria.

## Abstract

Models of consumer effects on a shared resource environment have helped clarify how the interplay of consumer traits and resource supply impact stable coexistence. Recent models generalize this picture to include the exchange of resources alongside resource competition. These models exemplify the fact that although consumers shape the resource environment, the outcome of consumer interactions is context-dependent: such models can have either stable or unstable equilibria, depending on the resource supply. However, these recent models focus on a simplified version of microbial metabolism where the depletion of resources always leads to consumer growth. Here, we model an arbitrarily large system of consumers governed by Liebig's law, where species require and deplete multiple resources, but each consumer's growth rate is only limited by a single one of these resources. Resources that are taken up but not incorporated into new biomass are leaked back into the environment, possibly transformed by intracellular reactions, thereby tying the mismatch between depletion and growth to cross-feeding. For this set of dynamics, we show that feasible equilibria can be either stable or unstable, again depending on the resource environment. We identify special consumption and production networks which protect the community from instability when resources are scarce. Using simulations, we demonstrate that the qualitative stability patterns derived analytically apply to a broader class of network structures and resource inflow profiles, including cases where multiple species coexist on only one externally supplied resource. Our stability criteria bear some resemblance to classic stability results for pairwise interactions, but also demonstrate how environmental context can shape coexistence patterns when resource limitation and exchange are modeled directly.

## Author summary

One longstanding challenge in community ecology is to understand how diverse ecosystems assemble and stably persist. Microbial communities pose a particularly acute example of this open problem, because thousands of different bacterial species can coexist in the same environment. Interactions between bacteria are of central importance across a

**Funding:** This material is based upon work supported by the National Science Foundation Graduate Research Fellowship Program under Grant No. DGE-2039656 and Grant No. DGE-1746045. T.G. was supported by Grant No. DGE-2039656 and Z.R.M. was supported by Grant No. DGE-1746045. Any opinions, findings, and conclusions or recommendations expressed in this material are those of the author(s) and do not necessarily reflect the views of the National Science Foundation. J.P.O. acknowledges funding from Simons Foundation Grant No. 376199 (www.simonsfoundation.org) and McDonnell Foundation Grant No. 220020439 (www.jsmf.org). The funders had no role in study design, data collection and analysis, decision to publish, or preparation of the manuscript.

**Competing interests:** The authors have declared that no competing interests exist.

wide variety of systems, from the dynamics of the human gut microbiome to the functioning of industrial bioreactors. As a result, a predictive understanding of which microbes can coexist together, and how they do it, will have far-reaching applications. Here, we incorporate a more realistic understanding of microbial metabolism into a classic mathematical model of consumer-resource dynamics. In our model, bacteria deplete multiple abiotic nutrients but their growth rates are only sensitive to one of these resources at a time. In addition, they recycle some of the nutrients they consume back into the environment as new (transformed) resources. We analytically derive criteria which guarantee that any number of microbes will coexist. We find that there are special types of interaction networks which remain stable even when resources are scarce. Our theory can be used in conjunction with experimentally determined interaction networks to predict which species assemblages are likely to stably coexist in a specified resource environment.

## Introduction

Pairwise interaction models have informed our understanding of when competitive interactions will lead to stable equilibria. For example, these classic models imply the coexistence of two competing species when the strength of interspecific competition is less than the strength of intraspecific competition, as well as more general stability criteria for large, multi-species systems with randomly distributed interaction strengths [1–4]. On the other hand, models of pairwise interactions do not explicitly include the effect of environmental context, and this context has the potential to refine or modify our understanding of when a group of interacting species will coexist. For example, one species may exclude another if both compete for and rely on a given resource, but the same two species may coexist if that resource is replaced by two alternative resources, each of which is consumed by only one of the two species.

Recent consumer-resource models incorporating the exchange of resources alongside resource competition have shed light on stable coexistence in systems where interactions are mediated by abiotic resources [5–11]. In these open systems, the environmental context is specified by resource inflow from outside, and stability turns out to depend both on the structure of which species consume and produce specific resources, and on the resource inflow rates. However, this recent theory has focused on a simplified version of microbial metabolism where the depletion of resources always leads to consumer growth. Specifically, the models studied in [5] assumed either that the impact of consumers *on* resources was proportional to the growth rate resulting from consumption of these substitutable resources (ie. the impact of resources on consumers), or else a generalization of this type of assumption [6, 12, 13] for multiplicative colimitation by essential resources.

In general, the rates at which a consumer depletes resources are not always proportional to the benefits that it derives from their consumption. This kind of mismatch between impact and growth can arise for many reasons. One example is 'waste' by consumers with large uptake rates [14], where otherwise usable resources are degraded and made unavailable for other consumers. An even more basic origin for this mismatch arises when internal metabolism requires multiple resources, but is only limited by the availability of a single resource—known as Liebig's law [15–19]. In this case, a consumer will deplete and make use of multiple resources, but its growth rate will only be sensitive to one of those resources at a time. Thus, a given consumer can strongly affect the growth rate of others that *are* limited by one of its own non-limiting resources.

The consumption of non-limiting resources does not result in biomass production or cell division. These non-limiting resources could instead be used for cellular maintenance, or transformed via cellular metabolism into byproducts and then leaked back into the environment [20–22]. The depletion of non-limiting resources is one way to generate a mismatch between consumption and microbial growth. Because of the conservation of resource biomass, this mismatch gives rise to the production of resource byproducts, and hence the potential for cross-feeding. For example, a recent reconstruction of the metabolic evolution of the marine cyanobacteria *Prochlorococcus* suggests that *Prochlorococcus* is nitrogen limited and leaks organic carbon, forming a mutualism with the heterotrophic bacterium SAR11 [14]. Both the mismatch between depletion and growth as well as the cross-feeding of nutrients may have important ecological consequences because they shape the resource environment for competing species. Yet, we do not know whether this more realistic picture of microbial metabolism changes our theoretical understanding of coexistence in diverse microbial communities.

In this paper, we model an arbitrarily large system of consumers undergoing growth governed by Liebig's law. We consider dynamics near a positive equilibrium where each consumer is limited by a single, distinct resource, but can potentially deplete additional resources. Each consumer leaks the resources it does not use for growth back into the environment in other forms, thereby tying the mismatch between growth and depletion directly to cross-feeding. Although we are primarily modeling microbial communities, the mismatch between growth and depletion is a general ecological phenomenon, and our results apply equally well to non-microbial systems. We find that with certain additional assumptions it is possible to analytically derive sufficient stability criteria in terms of resource inflow rates and ecological network structure which guarantee that a feasible equilibrium is stable. These criteria mirror those found earlier for a different form of positive interactions [5], and show that the structure of consumption and production networks, as well as the environmental context, affect the stability of this equilibrium. Our theory generalizes well-known results for low diversity consumer-resource dynamics [23], but also identifies stabilizing interaction network structures which do not have a clear low-dimensional analog. Using simulations, we show that our stability criteria apply more broadly to network structures and parameter regimes which do not precisely satisfy our mathematical assumptions, including situations where many microbial species coexist on only one externally supplied resource [9, 10]. As a result, our theory could be used to select species assemblages whose consumption preferences and nutrient production networks permit coexistence in a specified resource environment.

## Materials and methods

### A model of the consumption and cross-feeding of resources

We consider a model with five basic biological processes—resource supply, consumption of resources, consumer growth, consumer mortality and cross-feeding. In Fig 1A, we illustrate how these processes shape the flow of resources into and out of a focal consumer, while in Fig 1B, we show the resulting flow of resources in the community. To develop a mathematical description of these dynamics, let $\vec{R}$ (respectively $\vec{N}$) be a vector of $S$ resource (respectively consumer) abundances. Resources are externally supplied and depleted through consumption. Let $\rho_i$ be the inflow rate of the $i$-th resource, and let $C_{ij}$ be the rate of consumption of resource $i$ by consumer $j$. We will denote the vector of $\rho_i$ values by $\vec{\rho}$ and the $S \times S$ matrix of consumption rates by $C$. We also define $\epsilon_{ij}$ to be the efficiency at which consumer $j$ converts resource $i$ into new biomass, and we collect these parameters into an $S \times S$ matrix $\epsilon$ with values in the interval [0, 1]. Intuitively, these parameters are inversely related to the stochiometric requirements of each consumer on each resource. If a given consumer has a larger requirement for one

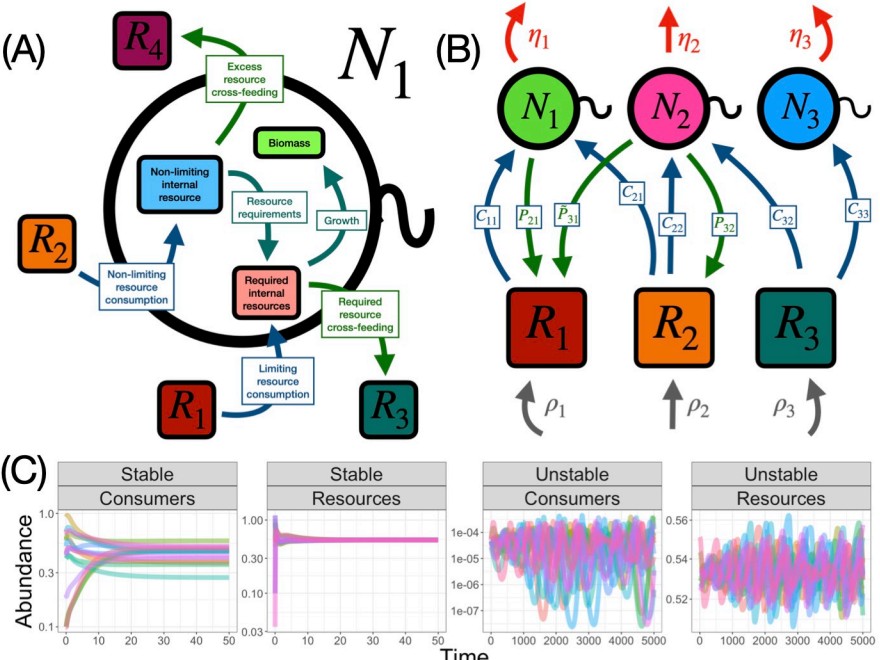

**Fig 1. Conceptual diagrams of internal metabolism, resource flow in the model community, and example consumer-resource dynamics.** (A) A schematic of the internal metabolism in our model for one consumer $N_1$. This consumer depletes two resources—$R_1$ is limiting for consumer $N_1$ while $R_2$ is not. The consumption of resource $R_1$ goes fully towards towards the consumer's internal requirements. Some portion ($\theta$) of this consumption is leaked out into the environment (as resource $R_3$ in the diagram), while the remaining consumed resource $R_1$ goes towards biomass. Because resource $R_2$ is not limiting, it is consumed in excess of consumer $N_1$'s requirements for growth. This excess consumption is recycled back into the environment (as resource $R_4$). The remaining resource then goes to biomass producing processes, exactly as the entirety of the limiting resource intake. Therefore, some of the required consumption of resource $R_2$ is leaked and the rest goes to biomass. (B) A schematic of the consumer-resource model in Eq (1). Consumers ($N_i$) deplete the resource ($R_j$) at rates $C_{ij}$ (all blue arrows), but only some of the consumption goes to consumer growth. Instead, some consumed resource is leaked back into the environment as new resources at rates $P_{ji}$ or $\tilde{P}_{ji}$ (green arrows). Resources are externally supplied at rates $\rho_i$ (gray arrows) and consumers undergo density independent mortality at rate $\eta_k$ (red arrows). (C) The dynamics of the consumers and resources in Eq (1) when there is a stable equilibrium and when there is not. Consumers grow according to Liebig's law as described in the text. When there is no stable equilibrium, the consumer abundances undergo large fluctuations, reaching low abundances.

resource relative to a second resource, it will have a smaller efficiency parameter for this first resource, and therefore it must consume more of the first resource than the second to build the same amount of biomass.

We assume that each consumer requires all the resources that it depletes for growth. In other words, $C_{jk}$ is non-zero for these essential resources, and zero otherwise. Under Liebig's law, each consumer's growth rate is determined only by the resource which is least available relative to the consumer's growth requirements. A given consumer still builds biomass from all of these resources, but they are used in fixed proportions, so one resource limits the rate at which all others can be used to build biomass. Therefore, the growth rate of species $k$ is proportional to $g_k = \min_{j \in \{1, \ldots, S\}}\{\epsilon_{jk} C_{jk} R_j : C_{jk} \neq 0\}$ where we have required that $C_{jk} \neq 0$ so that the minimum is only over those resources that are necessary for consumer $k$. Throughout this work, we consider cases where all or only a subset of resources are needed for the growth of each consumer. However, this difference is not essential, and our main results apply equally well to any growth rate function which selects a single resource. For example, if we replace the minimum in the growth function by a maximum, we can interpret the resulting dynamics as a

simple model of preferential resource utilization, albeit in a different form than the classic understanding of diauxic shifts [24–27]. In addition to growth, consumer $k$ undergoes density independent mortality at rate $\eta_k$. We primarily focus on the case where $\eta_k = \eta$ and interpret this parameter as a constant washout rate, as in a chemostat [28]. In our model, resources are not washed out of the system, so we are implicitly assuming that all resources are quickly depleted by consumers [7, 29].

Lastly, consumers recycle nutrients back into the environment by cross-feeding. We consider two different types of cross-feeding in our model. First, each consumer depletes some resources that are non-limiting for it, and the excess intake of these resources is cycled back into the environment, possibly following some internal reactions that transform it. Mathematically, the amount of resource $j$ used by consumer $k$ for growth is given by $\frac{\epsilon_{k'k}}{\epsilon_{jk}} C_{k'k} R_{k'}$, where $k'$ is the index of the limiting resource for consumer $k$. The additional efficiency factor $\frac{\epsilon_{k'k}}{\epsilon_{jk}}$ converts the consumption of the limiting resource into the units of resource $j$. In other words, we assume that resources $j$ and $k'$ are used in the fixed ratio $\frac{\epsilon_{k'k}}{\epsilon_{jk}}$ to build new biomass. The excess consumption of resource $j$ is then given by $C_{jk}R_j - \frac{g_k}{\epsilon_{jk}} = C_{jk}R_j - \frac{\epsilon_{k'k}}{\epsilon_{jk}} C_{k'k}R_{k'}$. Before being secreted, this internal excess may be converted to other forms through intracellular reactions that do not create new biomass. These conversion processes are summarized by the matrix $P$, with $P_{ji}$ giving the fraction of consumed resource $j$ that is recycled back into the environment as resource $i$. In our model, each resource is transformed independently to produce a set of byproducts, although more complex internal reaction dynamics, where multiple consumed resources react to form byproducts together, are certainly possible [12]. Consumers in our model also leak some byproducts from the processes that convert resources to biomass. We interpret this leakage as an inefficiency due to the permeability of cell membranes, but there may also be evolutionary reasons why microbes secrete metabolites [30]. We assume that constant fractions, $\theta$, of the resources used to build biomass are transformed into byproducts, as described by another matrix $\tilde{P}$, which can be different from $P$. To ensure conservation of biomass, we require that $\sum_i P_{ji} = \sum_i \tilde{P}_{ji} = 1$. As a result, the system as a whole is always competitive, because the overall production of resources is bounded by their total consumption, even though there can be net flows of one resource to another. For tractability, we assume that the chemical reactions underlying resource transformation are universal, so that resource conversion rates are the same for every consumer.

All together, the dynamics of resources and consumers are given by

$$\begin{aligned}
\dot{R}_i &= \rho_i - R_i\sum_j C_{ij}N_j + \sum_j P_{ji}\sum_k \left(C_{jk}R_j - \frac{g_k}{\epsilon_{jk}}\right)N_k + \theta\sum_j \tilde{P}_{ji}\sum_k \frac{g_k}{\epsilon_{jk}}N_k \\
\dot{N}_k &= N_k((1-\theta)g_k - \eta_k) = N_k\left((1-\theta)\min_{j\in\{1,\dots,S\}}\{\epsilon_{jk}C_{jk}R_j : C_{jk}\neq 0\} - \eta_k\right)
\end{aligned} \qquad (1)$$

when the consumers grow according to Liebig's law. In Table 1, we define all the relevant variables in Eq (1) and those that we define later for ease of reference. We focus on the equilibria of this model. Specifically, we determine when the equilibrium of Eq (1) is robust to small perturbations by finding sufficient conditions for the eigenvalues of its Jacobian to all have negative real parts. In Fig 1C, we illustrate two different examples of the dynamics that can result. In one example, there is a stable equilibrium to which the consumer and resource abundances converge, while in the other, both consumers and resources undergo large fluctuations to very low abundances. In the following sections, we derive criteria for these equilibria to exist and be

**Table 1. Variables in Eq 1 and in our stability criteria with their meanings.**

| Glossary of mathematical variables | |
|---|---|
| $R_i$ | Abundance of resource $i$ |
| $N_k$ | Abundance of consumer $k$ |
| $\rho_i$ | Inflow rate of resource $i$ |
| $C_{ij}$ | Consumption rate of resource $i$ by consumer $j$ |
| $P_{ij}$ | Production rate of resource $i$ from resource $j$ via non-limiting resources |
| $g_k$ | Growth rate of consumer $k$ |
| $\epsilon_{jk}$ | Growth efficiency of consumer $k$ on resource $j$ |
| $\tilde{\epsilon}_{jk}$ | Efficiency conversion ratio for consumer $k$ on resource $j$ ie. $\tilde{\epsilon}_{jk} = \epsilon_{kk}/\epsilon_{jk}$ |
| $\theta$ | Fraction of consumed and limiting resources that is leaked |
| $\tilde{P}_{ij}$ | Production rate of resource $i$ from resource $j$ via limiting resources |
| $\eta_k$ | Death (or washout) rate of consumer $k$ |
| $B_{ij}$ | Net effect of consumer $j$ consumer $i$'s limiting resource |
| $C_d$ | Consumption of the limiting resource for all consumers |
| $r$ | Resource abundances when they are identical |
| $n$ | Consumer abundances when they are identical |

stable. Specifically, we show that the unstable dynamics in Fig 1C can be caused by a variety of factors, including the structure of the ecological network and insufficient resource supply.

## Equilibria of the consumer-resource model

At an equilibrium of the dynamics in Eq (1) where all of the microbial species coexist, each resource is limiting for precisely one consumer. If instead two consumers were limited by the same resource, whichever species could drive the limiting resource level to the lower level would exclude the other, consistent with the competitive exclusion principle [31–33]. Thus, close to equilibrium, the minimum of the growth rule $g_k(\vec{R})$ must be realized by a unique resource for each consumer, and there is a one-to-one correspondence between consumers and resources. In models with different forms of resource competition, for example, those where consumers can be co-limited by different resources [13], this type of one-to-one correspondence may not occur. Nevertheless, there is a natural assignment of resources to consumers because of the structure of the model we consider here. By re-ordering the columns of $C$ so that its diagonal corresponds to the limiting consumer-resource pairs, we can rewrite the dynamical system in (1) as

$$
\begin{aligned}
\dot{R}_i &= \rho_i - R_i\sum_j C_{ij}N_j + \sum_j P_{ji}\sum_k (C_{jk}R_j - \tilde{\epsilon}_{jk}C_{kk}R_k)N_k \\
&\quad + \theta\sum_j \tilde{P}_{ji}\sum_k \tilde{\epsilon}_{jk}C_{kk}R_k N_k \\
\dot{N}_k &= N_k((1-\theta)\epsilon_{kk}C_{kk}R_k - \eta_k)
\end{aligned}
\tag{2}
$$

because the limiting consumer-resource assignments no longer vary in time. Here we have defined the matrix of ratios $\tilde{\epsilon}_{jk} = \epsilon_{kk}/\epsilon_{jk}$. In Fig A of the S1 Appendix, we plot the dynamics of models in Eqs (1) and (2). After a transient period in which the dynamics of these two models differ, both converge to the same equilibrium, demonstrating that the abundances need not be infinitesimally close to equilibrium for the simplified model to apply. More generally, whenever we find an equilibrium of the more complex model in Eq (1), we can find a corresponding model in the form of Eq 2 with the same equilibrium properties. Therefore, we restrict our

attention to the stability properties of Eq (2) because this model captures the behavior of models with more biologically realistic growth rules in a neighborhood of equilibrium.

Before we derive stability criteria for the equilibria of the dynamics in Eq (1), we must determine whether or not there are equilibria of the model in the first place. Specifically, we want to characterize when there are equilibrium solutions where all species coexist at positive abundance (called feasible equilibria [34]). Let's first analyze the simplified system in Eq (2). The resource abundances at equilibrium are immediately determined by the consumer dynamics to be $R_i^\star = \frac{\eta_i}{(1-\theta)\epsilon_{ii}C_{ii}}$. The equilibrium abundances of the consumers are given by

$$\vec{N}^\star = [(I - P^T)\vec{R}_d^\star C + (P^T - \theta\tilde{P}^T)\tilde{\epsilon}\vec{C}_d\vec{R}_d^\star]^{-1}\vec{\rho} \tag{3}$$

where $\vec{R}_d^\star$ denotes the diagonal matrix with entries given by $R_i^\star$. When each of the consumer abundances is greater than zero ($N_i > 0$), the equilibrium is feasible and it is possible for all species to coexist at abundances that do not change over time. The feasibility of a given set of equilibrium abundances $\vec{N}$ depends on both the consumption and production matrices, as well as the resource inflows and washout rates. In general, it is a difficult problem to characterize the set of feasible abundances in terms of the other parameters of our model. Moreover, depending on the growth rule of Eq (1), the feasibility of the resource abundances may introduce additional constraints on the interaction patterns. For example, if consumers grow according to the Liebig's law growth rule described in the previous section, then there is an upper limit on how large the diagonal coefficients of $C$ can be before at least one consumer becomes limited by a different resource at equilibrium. We use a combination of theory and simulation to ensure that we are analyzing the stability of equilibria that are actually part of the dynamics in Eq (1).

## Sufficient stability criteria for constant abundances

For the sake of analytical tractability, we focus on a specific parameterization of the model in Eq (1) in this section. We choose efficiency ($\epsilon$) and washout ($\vec{\eta}$) parameters so that all resources have equal abundance ($\vec{R}^\star = r\vec{1}$). We assume that all consumers have equal abundance ($\vec{N}^\star = n\vec{1}$) by choosing a corresponding resource inflow vector $\vec{\rho}$. Later, we relax these assumptions in simulations and show that our qualitative stability results still apply. Throughout this paper, we take the diagonal consumption coefficients to be the same for each consumer ($C_{ii} = C_d$), implying that all species are equally good consumers of their limiting resource. Under these conditions, we state two criteria which, if satisfied, imply that the equilibrium where all species coexist is stable. These criteria are *sufficient*, but not necessary, for the stability of the entire community.

1. The matrices $C$, $P$ and $B = -C + P^T(C - C_d\tilde{\epsilon}) + \theta C_d\tilde{P}^T\tilde{\epsilon}$ are symmetric.

2. All of the eigenvalues of $B$ are negative.

We derived these conditions through a mathematical analysis of the Jacobian of the equilibrium with constant abundances, which we detail in the Stability Criteria section of the S1 Appendix. Mathematically, the matrix $B$ is the upper right-hand block of the Jacobian, which measures how the resource abundances change as a function of the consumer abundances. Interestingly, it also appears in Eq (3)—a connection we explore further in the Sufficient Stability Criteria Imply Feasibility section of the S1 Appendix.

The first criterion enforces the reciprocity of interactions between the consumers in the ecosystem. The symmetry of the consumption matrix $C$ requires each consumer to deplete the limiting nutrients of the other consumers in the system in exactly the same way that other

 

consumers deplete its limiting nutrient. The symmetry of the production matrix $P$ has a similar interpretation, but for resource production. When the production matrix is symmetric, cross-feeding of resource $i$ produces exactly the same amount of resource $j$ as resource $j$ produces of resource $i$ when it is cross-fed. The matrix $B$ acts as an effective interaction matrix for the community. The matrix element $B_{ij}$ is the net effect that consumer $j$ has on resource $i$, since it incorporates both the consumption of resource $i$ by consumer $j$ and all of the ways that consumer $j$ produces resource $i$ via cross-feeding. When the interaction matrix $B$ is symmetric, consumer $j$ alters the dynamics of resource $i$ in the same way that consumer $i$ alters the dynamics of resource $j$. Since each consumer's growth is determined by only one resource, we can interpret the symmetry of interactions in $B$ as perfectly balanced pairwise competition for each limiting resource. Consumer $i$ is limited only by resource $i$, so consumer $j$'s effect on resource $i$ (and hence consumer $i$'s growth) is exactly matched by consumer $i$'s effect on resource $j$ (and hence consumer $j$'s growth). Although we don't expect natural systems to be precisely symmetric, these symmetric structures represent an interesting limiting case for our analysis. Moreover, previous work has found that interaction networks are stabilized by being near to an exactly reciprocal structure [5, 6, 35]. In the Results, we demonstrate the same behavior—as the interactions in the matrix $B$ becomes close to a special network structure with these properties, it is more easily stabilized.

The second criterion shows that the stability of the matrix $B$ informs the stability of the ecosystem as a whole. This finding reinforces our interpretation of $B$ as an effective interaction matrix for the community, which summarizes the dynamical contributions of all five biological processes around equilibrium. There are many ways that our model parameters could change to stabilize or destabilize the matrix $B$, but we focus on interrogating stability as we vary the consumption coefficients of the limiting resources ($C_{ii} = C_d$). The parameter $C_d$ controls the self-regulation in the system by determining the level of intraspecific competition for the consumers. As $C_d$ becomes large, each consumer regulates its own resource more strongly than the other consumers affect it, and $B$ is more likely to be stable. This mirrors classic results for consumer-resource models with two resources [23], negative feedback in the theory of plant communities [36, 37] and the central role of self-regulation in multi-species stability theory [2, 3]. The first criterion is not sensitive to the magnitude of the consumption or production coefficients, as long as they are arranged in such a way to produce a symmetric $B$ matrix. Because the second criterion measures the relative strength of the interactions, it does depend on the magnitude of the consumption and production coefficients.

We focus on two structures for consumption and production networks that give rise to symmetric $B$. In our first parameterization, we consider a case where the consumption matrix $C$ is a symmetric matrix with row (and column) sums that are all equal (see Fig 2A). We term this parameterization the tradeoff case, since all consumers share the same overall metabolic capability, as in [29]. In this parameterization, we also assume that each resource is equally recycled back into the environment as all the other resources (Fig 2B). In our second parameterization, we take the consumption matrix $C$ and the production matrices $P$ and $\tilde{P}$ to both be symmetric circulant matrices (see Fig 2C and 2D). In a circulant matrix, each row is composed of the same values, but the values are shifted one element right as we descend the rows. In this case, we effectively fix the distribution of consumption or production rates for all species, but vary the identity of which resources are consumed or produced more or less strongly between consumer species. In addition to these matrix parameterizations, which generate symmetric $B$ matrices and therefore satisfy our first criterion, we also consider a parameterization which explicitly violates our first condition. For this parameterization, which we call the unstructured case, we simply sample the coefficients of $C$, $P$ and $\tilde{P}$ identically and independently from

 

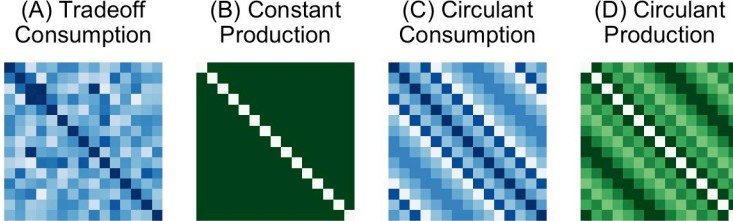

**Fig 2. Example consumption and production structures.** Each panel is a visualization of the different consumption and production patterns which satisfy our analytical criteria. Darker colors indicate larger values. (A) The tradeoff parameterization of the consumption matrix $C$. $C$ is symmetric with identical row (or column) sums and diagonal coefficients all set to $C_d$. (B) The constant parameterization of the production matrix $P$. Each entry off-diagonal entry is set to $\frac{1}{S-1}$, while the diagonal entries are set to 0. (C) The circulant parameterization of the consumption matrix. Each row of the matrix is a permuted version of the previous row and the diagonal entries are all set to $C_d$. (D) The circulant parameterization of the production matrix. Each row of the matrix is a permuted version of the previous row and the diagonal entries are set to 0.

probability distributions. Even though our first stability condition will not hold in this case, our second stability condition could still be satisfied. Therefore, we can test whether or not the first sufficient stability criterion is actually necessary to ensure stability, or if just the second criterion will suffice. In the Matrix Parameterizations section of the S1 Appendix, we list additional matrices that were studied, as well as more detailed descriptions of how we generate each case numerically.

## Results

To evaluate how well our sufficient stability criteria predict the onset of stability in simulated ecosystems, we generated large numbers of random matrices according to each of our parameterizations. We then numerically computed the smallest diagonal consumption $C_d$ value at which the coexistence equilibrium becomes stable. For this first set of results, we do not test whether or not the equilibrium is feasible, so that we can directly evaluate our stability theory. In the tradeoff and circulant cases, however, we show that feasibility is guaranteed when the consumer (respectively resource) abundances are all equal, (see the Feasibility Analysis section of the S1 Appendix for the proof). As a result, only the empirical $C_d$ values for the unstructured parameterization could be affected by feasibility constraints, and we describe the feasibility of this unstructured parameterization in detail in the Feasibility Analysis section of the S1 Appendix. In Fig 3A–3C, we plot our sufficient analytical stability bound against the numerically-determined $C_d$ values which first induce stability. For all three parameterizations, the empirically computed $C_d$ values are closely predicted by our analytical $C_d$ values. The fact that the unstructured case can ever be stable demonstrates that our stability criteria are not necessary for stability, since this case violates our first symmetry condition. In Fig 3D–3F, we plot the differences between the predicted and empirically-observed stability bounds. For the cases which satisfy our first symmetry condition, these differences are due to numerical error and therefore effectively zero (Fig 3D and 3E), while in the unstructured case, the predictions remain quite accurate but are no longer exact (Fig 3F).

In our mathematical theory, we have only proven that our stability criteria are sufficient for, rather than exactly predictive of, stability. And yet, in Fig 3 and Fig C of the S1 Appendix, our theory accurately predicts the precise value of $C_d$ at which the system becomes stable. These simulations suggest a more comprehensive result—if $B$ is symmetric, then the equilibrium is stable if and only if $B$ is stable. Intuitively, this result suggests that, if the effective ecological interactions in the system obey a specific symmetric structure, then they are exactly

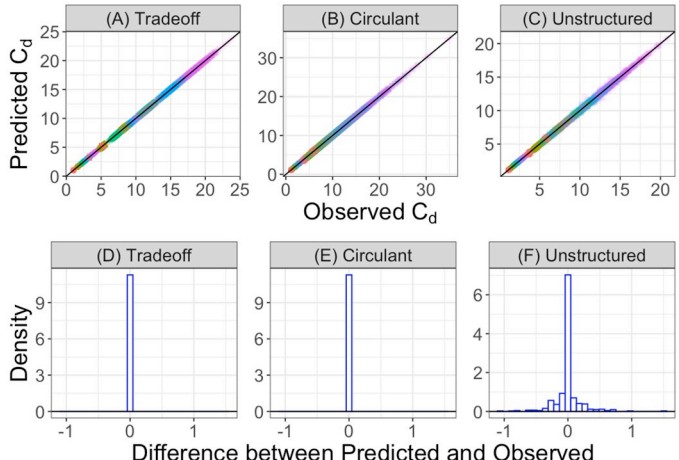

**Fig 3. Analytical stability criteria predict empirical values.** (A-C) We plot the value of the theoretical bound for $C_d$ in the second stability criterion against the smallest $C_d$ value at which the fixed point becomes stable numerically for 100 realizations of the random matrices comprising $C$, $P$ and $\tilde{P}$ while varying the standard deviation (colors) and mean (shapes) of the off-diagonal elements of the consumption matrix. The different panels are for the three different parameterizations of the consumption and production matrices ((A) is the tradeoff parameterization, (B) is the circulant parameterization and (C) is the unstructured parameterization). Parameters: $S = 15$, $n = r = 1$, $\theta = 0.5$, $\epsilon_{ii} = 0.05$ and $\epsilon_{ij} = 1$ for $i \neq j$ in all panels. For each of the different matrix parameterizations, we first sample the consumption coefficients from uniform distributions with mean given by the average consumption value (1, 3, or 5 in these simulations) and the specified standard deviations so that the coefficients of variation in consumption coefficients vary from $0.001/\sqrt{3}$ to $1/\sqrt{3}$. Then, we impose the constraints for the tradeoff and circulant parameterizations afterwards. (D-F) Histograms of the differences between the predicted and observed values in panels A-C in each of the three parameterizations when the average consumption coefficient is 5 and the coefficient of variation in consumption coefficients is $1/\sqrt{3}$.

predictive of the stability of the community as a whole. While we cannot prove this statement in its full generality, we present additional analytical arguments and numerical evidence in support in the Stability Criteria section and Fig J of the S1 Appendix. In summary, our first main result is that, when the first stability condition (symmetry) is satisfied, our second stability condition is sufficient for, and apparently exactly predictive of, the stability of the entire community. By contrast, in the unstructured parameterization, our analytical criteria do not exactly predict stability (Fig 3F). At the same time, our analytical bound predicted the *average* stability properties for the unstructured consumption and production matrices remarkably well (Fig 3C), even though our theory is not mathematically justified in this case.

Motivated by this surprising relationship, we next simulate our model more exhaustively to identify parameter combinations where our analytical criteria fail to predict the average behavior of the empirical $C_d$ values. We find that varying the consumer abundance $n$ causes the unstructured case to systematically violate our second stability criterion (see Fig D and E the S1 Appendix for the effect of other parameters). Our theoretical criteria exhibit no dependence on the value of $n$, and neither do the empirically computed $C_d$ values for the tradeoff or circulant cases (Fig 4A and 4B). Yet, we find that, as we decrease the consumer abundance $n$, it becomes increasingly more difficult for the equilibrium to be stable with unstructured consumption and production structures (Fig 4C). Our second main result is that the specific symmetric structure enforced by our first stability criterion protects the community from instability when $n$ is small, but, for non-symmetric consumption and production patterns, low consumer abundances lead to instability. This result mirrors other recent studies, which have

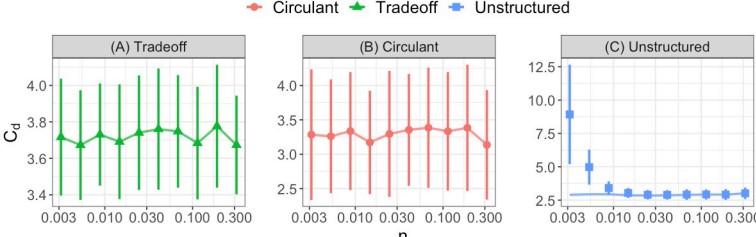

**Fig 4. Unstructured networks violate theoretical stability criteria at low consumer abundances.** We plot the value of the theoretical stability bound for $C_d$ averaged over 100 replicates (solid lines) as the consumer abundance $n$ varies over three orders of magnitude for the three different matrix parameterizations (panels and colors). We also plot the average smallest $C_d$ value for which the system first becomes stable numerically (shapes) along with error bars showing one standard deviation above and below the mean. In panels (A-B), the theoretical bound accurately predicts the dependence of $C_d$ on $n$, while for the unstructured case (panel (C)), the theoretical predictions fail at low consumer abundance. Parameters: $S = 15$, $r = 1$, $\theta = 0.5$, $\epsilon_{ii} = 0.05$ and $\epsilon_{ij} = 1$ for $i \neq j$. The off-diagonal elements of the consumption matrices are sampled from uniform distributions on $[0, 2]$ before the parameterizations are imposed.

also found that special network structures promote stability when consumer abundances are small as a result of low of resource supply [6, 7].

We now consider a new parameterization of our model that mimics recent microbial competition experiments [9, 10], where diverse communities coexist via cross-feeding. Above and in our analytical theory, we assume that all consumers and resources have equal abundances at equilibrium (ie. $\vec{N}^\star = n\vec{1}$ and $\vec{R}^\star = r\vec{1}$). We also treat these abundances as parameters that we can modify. In experimental microbial communities, the consumer and resource abundances ($\vec{N}$ and $\vec{R}$) cannot be fixed to specific values, but they can be manipulated indirectly by changing the resource inflow and consumer dilution vectors respectively. Using simulations, we now ask whether the qualitative stability patterns we have derived for the simpler case of constant abundances also describe systems where we treat $\vec{\rho}$ and $\vec{\eta}$ as independent variables and let the dynamics determine the consumer and resource abundances. To make contact with empirical scenarios, we set $\eta_i = \eta$ for all consumers, as in a chemostat, let $\rho_1 = \rho$, and set all other resource supply rates to zero ($\rho_i = 0$ when $i = 2, \ldots, S$).

In Fig 5A and 5B, we plot the fraction of ecosystems with a feasible and stable fixed point as we vary the total resource inflow $\rho$ and the diagonal consumption coefficient $C_d$ for the tradeoff and unstructured matrix parameterizations. We consider two different aspects of feasibility in these simulations. First, there must be a positive equilibrium which solves Eq (3). In other words, there must be a fixed point where all consumers coexist. Second, we require that the feasible fixed point found using Eq (3) is also realized in the dynamics of the Liebig's law growth rule. These two constraints impose lower and upper limits (minimum and maximum $C_d$) on the coexistence regions in Fig 5. Both of these feasibility properties, however, are independent of the total resource availability. Decreasing the resource inflow $\rho$ decreases consumer abundances, but it does not affect whether or not there is an equilibrium with all positive abundances. When the consumption and production matrices are unstructured, these low consumer abundances give rise to unstable, but feasible, fixed points (Fig 5B). By contrast, the consumption and production networks which obey our first symmetric stability criteria are protected from resource-inflow-mediated instability even in this more complex case (Fig 5A and Fig G of the S1 Appendix)). Thus, our third main result is that the symmetric consumption and production networks we identified analytically also promote stability when consumer and resource abundances are not identical.

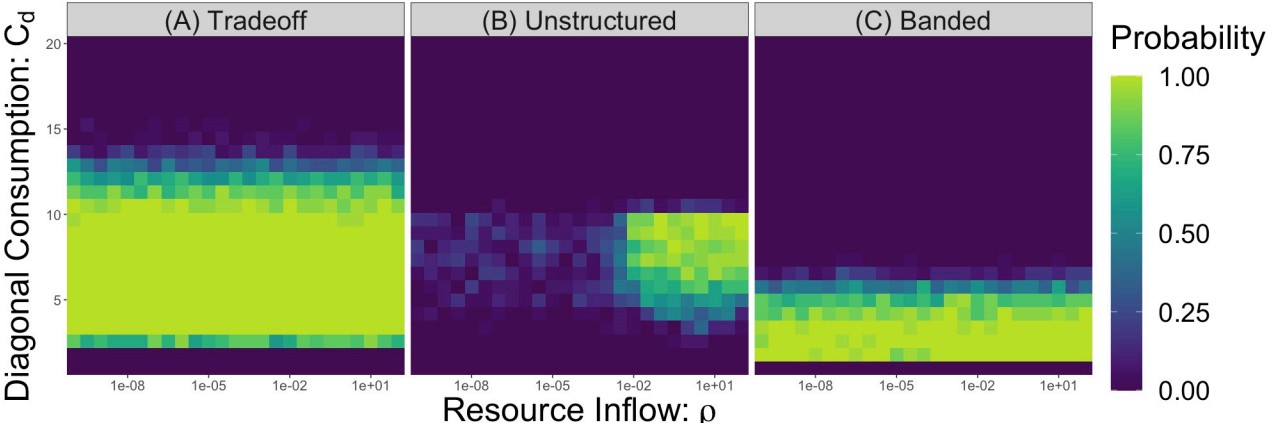

**Fig 5. Low consumer abundances induce instability for more general resource inflows and interaction networks.** We plot the probability of finding a feasible and stable fixed point in 25 replicates across a range of $C_d$ values and resource inflows $\rho$ with only one externally supplied nutrient for three different matrix parameterizations. We also enforce that the fixed point is realized under a Liebig's law growth rule, where each consumer grows on the most limiting nutrient of the resources. (A) The tradeoff matrix parameterization does not show any dependence on the resource inflow $\rho$, as in Fig 4. (B) The unstructured case does not have any feasible and stable fixed points at low resource inflow. (C) The banded matrix parameterization has a consumption matrix with non-zero values on the upper and lower bands of the matrix, displaced from the diagonal by one index. It also has a constant production matrix, as described previously. It does not show any dependence on the resource inflow. Parameters: $S = 15$, $\eta_i = 1$, $\theta = 0.9$, $\epsilon_{ii} = 0.05$ and $\epsilon_{ij} = 1$ for $i \neq j$. Consumption coefficients sampled from uniform distributions on [0.5, 1.5] before the constraints are imposed.

We examine this resource-inflow-mediated instability in a variety of other non-symmetric consumption and production networks to better understand the network properties that protect against it. We hypothesize that the reciprocal interaction structure of the $B$ matrix protects ecosystems from instability because it ensures that the $B$ matrix has purely real eigenvalues, as has been observed in previous work [6, 35]. In Fig 5C, we consider a parameterization of the consumption and production networks that generates an interaction matrix $B$ which is no longer symmetric but does have real eigenvalues. There are many ways to generate such a $B$ matrix, but here we take a consumption matrix with non-zero values on the upper and lower bands of the matrix, displaced from the diagonal by one index, along with constant production matrices. Once again, the equilibrium where all consumers coexist remains stable at low consumer abundances, indicating a more general result—namely, if $B$ is stable and has purely real eigenvalues, then the the entire community is stable. We present further mathematical analysis and numerical evidence in support of this claim in the Stability Criteria section and Fig G of the S1 Appendix.

We don't expect natural systems to satisfy our restrictive symmetry condition or to necessarily generate $B$ matrices that have eigenvalues whose imaginary parts are exactly zero. At the same time, it is possible that natural systems are better protected from instability if they are near to one of the special interaction structures we have identified analytically. Using simulations, we find that the transition between stability and instability at low consumer abundances is a continuous one. As we increase the correlation between off-diagonal pairs ($C_{ij}$, $C_{ji}$) of an otherwise random and unstructured $C$ matrix, its eigenvalues lie closer and closer to the real line, and we observe that the ecosystem can coexist at smaller and smaller resource inflows (see Fig H of the S1 Appendix). Therefore, our results extend to cases in which the $B$ matrix does not have precisely real eigenvalues. Instead, as the magnitude of the imaginary parts of the eigenvalues of $B$ decreases so too does the minimum resource inflow level at which the community can coexist. Our fourth and final result is therefore more speculative—we

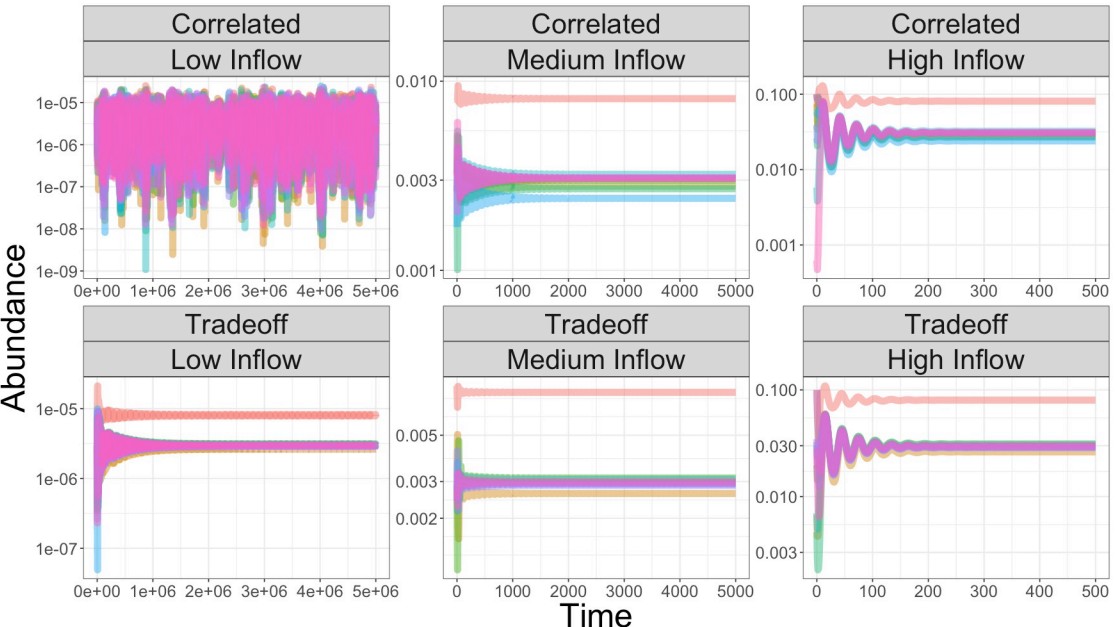

**Fig 6. Stable and unstable consumer dynamics for varying resource inflows.** We plot the consumer dynamics for two different consumption networks—one where the coefficients are sampled randomly from a distribution with correlations across the diagonal (labeled Correlated) and the other according to the tradeoff consumption network described in the text (labeled Tradeoff)—for three different magnitudes of one externally supplied resource (Low Inflow: $\rho_1 = 0.001$, Medium Inflow: $\rho_1 = 0.1$ and High Inflow: $\rho_1$ = 1). The $S = 15$ consumers' growth rates are determined by the most limiting resources following Liebig's law. All other resources are not supplied ($\rho_i = 0$ for $i = 2, \ldots, S$). In each case, the production network is given by the constant parameterization where all other resources are produced from a given incoming resource. For the tradeoff network, the equilibrium is stable regardless of resource inflow levels, while the correlated network becomes unstable when the resource inflow is low. Parameters: $S = 10$, $C_d = 10$, $\eta_i = 1$, $\theta = 0.9$, $\epsilon_{ii} = 0.05$ and $\epsilon_{ij} = 1$ for $i \neq j$. Consumption coefficients sampled from uniform distributions on [0.5, 1.5] before the constraints are imposed.

conjecture that the matrix $B$ serves as a measure of the relevant interactions in the ecosystem, and, when $B$ has all real and negative eigenvalues, the equilibrium is always stable. If $B$ has some eigenvalues with non-zero, but small, imaginary parts, the equilibrium will be stable across a broad range of resource inflows, and the threshold resource supply at which the system becomes unstable is partly determined by the magnitude of these imaginary parts.

Our theory predicts when a feasible and stable equilibrium does or does not exist in the dynamics of our model. It cannot, however, shed light on what happens to the consumers in an ecosystem without a stable equilibrium. In Fig 6, we plot the dynamics of two different consumption networks—a correlated but otherwise unstructured network and the tradeoff network–for three different inflow levels of one externally supplied resource. $B$ has purely real eigenvalues for the tradeoff consumption network, but some of the eigenvalues of $B$ have non-zero imaginary part for the correlated network. In the tradeoff parameterization, consumers converge to a stable equilibrium across the different resource inflows, while in the correlated parameterization, they converge to equilibrium only when the resource supply is sufficiently large. When the dynamics do not converge to a stable equilibrium, the consumers undergo large fluctuations which vary in amplitude (Fig 6). In the presence of demographic noise, these large fluctuations would likely lead to exclusion. In addition to these semi-regular fluctuations, consumers appear to converge to stable limit cycles in some cases. Depending on the initial conditions, it is also possible for some consumers to become limited by a different resource, leading to consumers reaching low abundances or even becoming excluded (see Fig K of the

S1 Appendix). In general, the unstable dynamics in our model can be highly irregular and lead to the exclusion of some consumers.

## Discussion

Unlike many classic ecological models in which species interactions are completely determined by the abundances of the competing species [38, 39], our model explicitly tracks the dynamics of resources [8, 40–42]. As the resource abundances vary over time, the per-capita effect of one consumer on another can change, in contrast to most pairwise models, where these per-capita interactions are fixed in time. Additionally, consumers grow according to Liebig's law, meaning that each consumer is limited by only a single resource at a time, even though they can deplete other resources. Liebig's law has been found to accurately describe microbial growth rates, as well as the resource limitation properties of plants, across many different ecosystems [12, 13, 18, 43–45]. In our model of Liebig's law, there is a mismatch between microbial growth and resource depletion that directly gives rise to the exchange of resources.

In this paper, we investigated how these two linked processes—microbial growth governed by Liebig's law and the subsequent cross-feeding of nutrients—jointly affect stability in arbitrarily large microbial communities. We analytically determined two stability criteria that are together sufficient for stability of the entire community. Our first stability criterion ensures that the interactions between the microbial species are reciprocal. Under our first condition, each consumer must affect the limiting resource of other consumers in exactly the same way that every other consumer affects its limiting resource. In our second stability criterion, we identified an effective interaction matrix for the community. When our first stability condition is satisfied, the stability of this effective interaction matrix implies that the equilibrium where all consumers coexist is also stable. Guided by these stability criteria, we found four main results. First, our two stability criteria were not only sufficient for, but in fact appear exactly predictive of, stability in simulated communities. Second, reciprocal interaction structures protected the community from instability at low consumer abundances, but communities with non-symmetric interactions generically became unstable when resources were scarce. Third, our analytical results applied qualitatively to situations where microbes compete for a single external resource and therefore have highly variable consumer abundances at equilibrium. Fourth, interaction networks that were not exactly symmetric still promoted stability, as long as the spectra of these networks had small imaginary parts.

Our model incorporates a more mechanistic picture of microbial interactions by considering how essential resources and cross-feeding alter ecological dynamics. At the same time, it is still a highly coarse-grained version of true microbial metabolism. We have focused on a particular form of Liebig's law with explicit stochiometric requirements, but other models without these requirements have been investigated as well [46–48]. In our consumption matrix parameterizations, we have also implicitly assumed that each consumer can deplete many of the available resources. Similarly, we have assumed that any resource can be produced from any other. In other words, although we have conserved total biomass in our model, we have not ensured that the basic biochemical building blocks are conserved when consumed resources are being converted into other nutrients. In reality, there are stochiometric rules that these matrices must obey [49–51]. Intriguingly, recent experiments [10] have shown that resource production networks are approximately, though not precisely, hierarchical because of biochemical constraints. Understanding how the chemical properties of abiotic nutrients and the metabolic strategies of specific bacterial strains constrain the consumption and production networks is therefore an important direction for future work. It would be particularly

interesting to evaluate whether these biochemical constraints on resource exchange create production networks with spectral properties similar to those we have shown to promote stability.

Even though our consumer-resource model does not capture the full complexity of microbial metabolism, it still produced stability criteria that refine our understanding of microbial coexistence. Recent theory [52, 53] has shown that population abundances do not affect stability in a randomly parameterized Lotka-Volterra model. In the present work, we find the opposite result—given consumption and production networks which do not satisfy our symmetry conditions, some choices of equilibrium consumer abundances generate stable systems, while others do not. By contrast, our theoretical results do align with other analyses of consumer-resource dynamics. The resilience of a food web (defined as the speed at which the abundances return to equilibrium after a perturbation) has been shown to increase as the residence time of nutrients in the system decreases [54]. One possible interpretation of our results is that, when resource supply is low, resources remain in the community for longer times, inducing instability, but further work is needed to understand this connection more completely. Recent simulations of model microbial communities with cross-feeding showed that, when consumers are resource limited, the constituent species interact in a characteristic pattern at equilibrium [7]. These results mirror our simulations, where ecosystems with specific symmetric interaction structures are protected from instability at low resource inflow. In [7], the characteristic interaction patterns emerge from community assembly, while in our theory, we impose them from the outset. This connection is particularly interesting because it suggests that special interaction structures may emerge from assembly processes in specific resource environments.

Similarly, a recent mathematical analysis of consumer-resource models with multiple forms of consumption also showed that resource inflow mediates a transition to instability [6]. This recent theory, however, treats resource exchange as coming directly from consumer biomass, as though resource production were an additional source of mortality, rather than as an explicit transformation of resources. As a result, the overall strength of production for each species is a tunable parameter and, if it exceeds the total consumption of a single species, then the feasible equilibrium can be unstable [6]. In our model, the strength of production is determined by the resource consumption that is not used for growth, and so cannot exceed the total consumption for each species. Nevertheless, we find that unstable equilibria are still possible due to the *mismatch* between resource depletion and consumer growth, rather than from the strength of resource production overwhelming resource depletion, as in [5].

Our results can also be seen as a multi-species generalization of classic stability results for species competing for two resources (termed contemporary niche theory) where instability occurs because of the difference between impact and sensitivity vectors [23, 55–57]. In contemporary niche theory, there are three criteria which must be satisfied for a stable equilibrium to exist. First, the species zero net growth isoclines (ZNGIs) must intersect. In our model, this is always true, since each species is limited by a single resource, so it is straightforward to find resource abundances where every consumers' growth is zero. Second, each species must impact the resources that it finds most limiting more strongly than it impacts other resources. Our second stability criteria is a direct and quantitative generalization of this result—if each species more strongly regulates the resource it requires for growth than it affects all other resources in the system, then the equilibrium is stable. Third, the supply point must lie above the ZNGIs for the coexisting species. In our theory, we ensure that this criteria is satisfied by requiring the equilibrium to be feasible through Eq (3). We also show in the Feasibility Analysis section of the S1 Appendix that our second stability criterion and the feasibility criteria are closely related—as species more strongly regulate their most limiting resource, the likelihoods of both stability and feasibility are increased. Our theory can be interpreted as an extension of

prior work applying contemporary niche theory to competition for essential nutrients to diverse microbial communities [23, 58].

By contrast, there is no clear analog of our first stability criterion for low diversity ecosystems. It can, however, be interpreted as requiring perfectly balanced pairwise competition between the consumers, even though the model itself is not built on pairwise competition coefficients. The phenomenon that reciprocity promotes stability has been found in other theoretical studies of microbial communities [6], but also in a diverse set of other fields, from the evolution of cooperation [59, 60] to the exchange of food in early societies [35, 61]. In addition to our symmetry condition, we showed numerically that other consumption and production networks that generate $B$ matrices whose eigenvalues are purely real also prevent instability at low consumer abundances. Although we lack a precise understanding of how these network structures promote stability, we describe intuitively how the imaginary parts of the eigenvalues of $B$ affect the spectrum of the Jacobian in the Stability Criteria section of the S1 Appendix. A more complete mathematical understanding of the connection between the spectrum of the interaction matrix $B$ and the spectrum of the Jacobian for the entire community would give us a deeper understanding of why certain modes of resource exchange are stabilizing.

There are a number of other important directions for future work. Previous simulations of consumer-resource dynamics governed by Liebig's law have found that large numbers of species can coexist in oscillatory or chaotic dynamics [46–48], so it would be instructive to better understand the behavior of our model away from equilibrium. Similarly, we have only observed a single unique equilibrium in our models, but cross-feeding can generate multiple equilibria [62]. Our stability criteria may still be able to delineate which of these multiple equilibria are attracting. It would also be interesting to rigorously understand the stability properties of an ecosystem where consumers grow on many different substitutable resources at variable efficiencies but still leak resources back into the environment through cross-feeding [7]. Last, the model we have considered here is completely deterministic, and so the mismatch between resource depletion and consumer growth resulted from a specific modeling choice. In a stochastic model, depletion and growth may be decoupled through only the differing fluctuations that the consumers and resources undergo, potentially yielding the same stability transition we have observed in our model. More generally, the combination of stochastic drift and the biological mechanisms we have explored here could produce interesting macroecological patterns [63, 64].

Because our consumer-resource model connects coexistence patterns to empirically accessible quantities, our theoretical results can be tested experimentally. One direct test of our theory would be to design small microbial communities with experimentally-characterized interaction networks [11, 65]. Then, an experimentalist can manipulate, for example, the degree to which the interaction network is reciprocal, and observe whether or not coexistence if favored. Conversely, our theoretical results suggest an interpretation of coexistence outcomes when the exact consumption and production networks are unknown. When an experimentalist reduces the resource inflow rates in a serial dilution experiment, the resulting coexistence (or lack thereof) suggests which types of consumption and production networks may be present. This relationship between the interaction network and the eventual stability properties of the entire community could also potentially be used to constrain the space of possible interactions when inferring these parameters from microbial abundance data [66–69]. Because we showed numerically that our analysis applies to a variety of resource inflow profiles, our theory may also delineate the boundaries of stable coexistence in recent experiments where only one nutrient is externally supplied [9, 10, 70]. For example, recent work has shown experimentally how the resource production network explains the variation in species richness as more resources are externally supplied [10]. Our theory suggests that, if this metabolite

production network has eigenvalues with small imaginary parts, the community will be better protected from resource scarcity. Future work should seek to further clarify how the relationship between coexistence outcomes and resource inflow changes depending on network structure. This line of research is especially important because of the difficulty in obtaining well-resolved and quantitative consumer-resource networks for diverse microbial communities.

## Supporting information

**S1 Appendix. Supplementary proofs, calculations and simulation results.** We provide proofs of the main results, additional numerical simulations and descriptions of all of the matrix parameterizations.
(PDF)

## Acknowledgments

We thank Seppe Kuehn, Simon Levin and Jonathan Levine for helpful comments and discussion. We also thank three reviewers for their insightful suggestions.

## Author Contributions

**Conceptualization:** Theo Gibbs, James P. O'Dwyer.

**Data curation:** Theo Gibbs.

**Formal analysis:** Theo Gibbs, Yifan Zhang, Zachary R. Miller, James P. O'Dwyer.

**Funding acquisition:** James P. O'Dwyer.

**Investigation:** Theo Gibbs, Yifan Zhang, Zachary R. Miller, James P. O'Dwyer.

**Methodology:** Theo Gibbs, Yifan Zhang, Zachary R. Miller, James P. O'Dwyer.

**Project administration:** James P. O'Dwyer.

**Software:** Theo Gibbs.

**Supervision:** James P. O'Dwyer.

**Validation:** Theo Gibbs.

**Visualization:** Theo Gibbs.

**Writing – original draft:** Theo Gibbs.

**Writing – review & editing:** Theo Gibbs, Yifan Zhang, Zachary R. Miller, James P. O'Dwyer.

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
