## [Decision Letter · Decision Letter 0]

3 Feb 2022

Dear Mr. Gibbs,

Thank you very much for submitting your manuscript "Stability criteria for the consumption and exchange of essential resources" for consideration at PLOS Computational Biology.

As with all papers reviewed by the journal, your manuscript was reviewed by members of the editorial board and by several independent reviewers. In light of the reviews (below this email), we would like to invite the resubmission of a significantly-revised version that takes into account the reviewers' comments.

The reviewers raised substantial criticism on the organization of the manuscript and its clarity, which makes it hard to judge the biological significance of the model assumption. If you decide to submit a revised version, it is important that you address all of the reviewers' concerns. This would make the manuscript much more accessible to the broad audience of Plos CB.

We cannot make any decision about publication until we have seen the revised manuscript and your response to the reviewers' comments. Your revised manuscript is also likely to be sent to reviewers for further evaluation.

Sincerely,

Jacopo Grilli

Associate Editor

PLOS Computational Biology

Natalia Komarova

Deputy Editor

PLOS Computational Biology

The reviewers raised substantial criticisms on the organization of the manuscript and its clarity, which makes it hard to judge the biological significance of the model assumptions. If you decide to submit a revised version, it is important that you address all of the reviewers' concerns. This would make the manuscript much more accessible to the broad audience of Plos CB

Reviewer's Responses to Questions

**Comments to the Authors:**

Reviewer #1: The paper extends consumer resource models with cross-feeding interactions to incorporate Liebig's law. The authors propose two stability criteria and find that the model exhibits chaotic behavior at low consumer abundances. I find the chaotic behavior very interesting, but I am very concerned about its biological significance.

1. The current form focuses on stability analysis or chaotic behavior, fitting more to a theoretical biology journal.

2. As noted by the authors, the stability and feasibility analyses require restrictive assumptions, which are hard to fulfill in real ecosystems. In the main results, eq. (3) ignores the difficulty of finding the feasible solution satisfying all consumer abundances positive. While eq. (4) is conditioned that all consumer abundances are identical, which requires resource supplied rates and the row sum of the consumer matrix are equal, respectively.

I can not recommend the paper to publish on PLOS Computational Biology for all the above reasons.

I also have some comments that could be useful if the authors decide to work more on the chaotic dynamics.

1. The authors do not distinguish between unstable and chaotic dynamics. The instability is the necessary condition for chaotic behaviors. Appendix 1.5 shows the theoretical proof for the case without cross-feeding. However, in the limit n->0, the resources undergo unbounded growth, which is unstable but not chaotic.

2. It would be helpful if the authors could clearly explain the origin of chaotic dynamics. I guess it results from random switches between different resource preferences, and thus eq. (2) should not exhibit chaotic behaviors. It is easy to check it numerically.

3. In principle, the population abundance can be rescaled to different values by tuning model parameters. If the chaotic behavior is general, it should not exhibit only “at low consumer abundance.” At least, it is possible to show the chaotic dynamics in order ~ 1. Otherwise, we cannot rule out the possibility of numerical instability in the ODE solver when the values are small.

Reviewer #2: This manuscript develops stability criteria for complex microbial communities that interact through resource competition and cross-feeding. Unlike previous related work, this study focuses on essential resources. The manuscript combines elegant analytical results from matrix theory with numerical simulations to flesh out the conditions for stability. Cross-feeding is common in microbes but relatively understudied, so this is a timely and important topic. I believe the manuscript will be ultimately publishable, but have some questions about the model formulation and suggestions to improve the clarity.

1) I was a bit unsure about the formulation of essential resources using Liebig's law of the minimum. In the usual interpretation, all resources contribute to production of new biomass in a fixed ratio. By definition, non-limiting essential resources don't LIMIT growth, but are still REQUIRED for growth. However the authors interpret this as "each consumer can only produce new biomass using one resource at a time ... we define the function g_k(R) to choose which resource the k-th consumer grows on" (lines 82-84). A consequence of this assumption is that whatever resources a consumer does not "grow on" have delta_jk=0, so in eq. 1 we see that they are completely excreted by the consumer. It seems that after excreting all of the non-limiting resources, none will remain for growth requirements so they actually should become limiting. Previous work on resource excretion takes into account this need for balance (see e.g. Sterner 1997 Freshwater Biol. 38: 473-481). Thus the formulation does not line up with how essential resources are commonly considered, unless I'm not following the argument.

2) The authors consider both stability and feasibility of coexistence equilibria, but not uniqueness. It is possible in cross-feeding interactions to have multiple coexistence equilibria, some stable, some unstable (Sun et al. 2019 JTB 465: 63-77). Therefore if one equilibrium is unstable, it does not necessarily mean that the species can't coexist. Is this a possibility in the authors' model?

The manuscript is complex, so the authors should do more to help readers. Here are some places I was confused. With sufficient effort I could figure these out, but I think the authors should make it easier on the reader.

3) In the statement of the two stability criteria in lines 177-178, it is not clear whether these two criteria BOTH need to be satisfied to guarantee stability or EITHER suffices. It's also not clear whether these are necessary or merely sufficient. Since this is a central result that structures much of the rest of the paper, this should be made obvious.

4) The description of choices of interaction matrices was confusing. Line 236 says that THREE interaction matrix structures will be studied, line 238 says we consider TWO different methods of parameterizing the matrices, see Fig. 2, which has FOUR matrices. This could use more careful exposition.

5) In writing my review, I thought I'd summarize the main findings, but I realized I wasn't sure what they were. The abstract says "We identify special consumption and production networks which protect the community from instability when resources are scarce". The discussion says "We identified a set of symmetric consumption and production networks that guarantee stability as long as intra-specific regulation is large compared to inter-specific interactions" (lines 475-477). What are those networks that promote stability? What are the key parameters that affect stability and how? Those should be the main conclusions, but they're not easy to find. The take-away messages should be made obvious in the abstract and highlighted in the beginning of the discussion.

Some other stylistic suggestions:

6) Make a table of key variables and be more liberal in using verbal descriptions of variables in the text.

7) Don't refer just to "S1 Appendix", give specific parts so readers can find the relevant section.

Reviewer #3: The authors model an arbitrarily large number of microbial species feeding on an equal number of resources. The chemostat model includes cross-feeding, or the use by one organism of metabolites released by other organisms. The authors find conditions for feasibility and stability of the system that depend on structural aspects of the system.

Comments

Figure 1. It appears that the red arrows mean that the consumer that die are removed from the system, as washout, and that is reflected by eta*N in equation (1). But there is no similar washout of the resources. Is that because uptake rates are assumed much higher than washout rates of resources?.

As the authors note, their model has some things in common with the well known models based on zero net growth isoclines (ZNGIs) and follow the general rules of those models regarding stability; each consumer being limited by one resource, each species regulating the resource that limits it more than any other resource, and the supply point being above the ZNGIs. I am guessing that the resources then are inorganic and/or organic compounds that enter the chemostat are taken up and either used to build biomass (only one of them) and the others released back into the environment. The authors make some additional assumptions.

“… each consumer can only produce new biomass using one resource at a time (which we define as the limiting resource) … This is a restrictive assumption, since some bacteria may incorporate biomass from non-limiting resources in nature.” (Lines 82-86)

The last sentence sounds like an understatement to me, as I would have assumed that in general bacteria would be taking up resources other than its limiting resource. Is there some empirical evidence that the assumption being made here is even approximately reasonable?

Lines 131-132. “each resource is limiting for precisely one consumer because of the competitive exclusion principle.” That perhaps shouldn’t be stated so absolutely, a co-limitation seems to be common, as least in primary producer communities; e.g. Harpole, W.S., Ngai, J.T., Cleland, E.E., Seabloom, E.W., Borer, E.T., Bracken, M.E., Elser, J.J., Gruner, D.S., Hillebrand, H., Shurin, J.B. and Smith, J.E., 2011. Nutrient co‐limitation of primary producer communities. Ecology letters, 14(9), pp.852-862. Clearly, of course, the assumption of one limiting resource for each consumer makes equation (2) and analysis much simpler.

Lines 187-190. “Consumer i grows only on resource i, so consumer j’s effect on resource i (and hence consumers i’s growth) is exactly matched by consumer i’s effect on resource j. Although we don’t expect natural systems to be precisely symmetric….” (This sentence is repeated almost word for word in lines 404-406.)

Again, the second sentence sounds like an understatement. I don’t see why that sort of symmetry, or reciprocity of interactions, should even approximately occur. If it does, it is very interesting, but it would be good to see at least some evidence that this might be roughly a pattern in nature. I realize this is necessary for matrix B to be symmetric, which then of course has the very nice property of negative eigenvalues. So it is an interesting idea that bacterial systems could evolve toward this sort of situation, where they are stable.

Line 204. “satisfying Eq. 4 ensures that each consumer regulates its own resource more strongly than other consumers affect it”

This calls to mind that another approach to large systems, in this case host-microbe systems, is that of James Bever and colleagues; e.g., Bever, Westover, and Antonovics 1997; Eppinga et al. 2018. The modeling of Bever and colleagues is different in that it deals with hosts (plants) and their microbial symbionts, which may be pathogens or mutualists, rather than consumers and resources in the present case. However, some aspects are similar, such that it deals with large systems, that overall self-limitation of individual species, or negative feedback at the system level leads to stability, and that there are two types of entities, unlike Lotka-Volterra competition models. A difference is that the Bever et al has models have actually been used successfully to interpret empirical data.

Line 221 “the second criterion ensures that the strength of the inter-species interactions is not too large relative to the inter-species interactions in the community”. This is somewhat trivial as it is well known. Why investigate this?

Lines 242-243. ‘so that each consumer produces an equal mix of all the other resources as it consumes resource i.” From Figure 1 resources other than i are leaked from the consumer after being taken up. Does the term ‘produces’ here mean that the consumers are making resources?

Lines 235-283. The authors present a number of different consumptions and production structures that satisfy the ‘first criterion’, or the reciprocity of interactions. They test if their analytic stability conditions work on these systems for different parameterizations. This is certainly interesting, but in the absence of presenting empirical evidence that microbial communities follow such reciprocity, there is not

Lines 417-418. “the threshold at which the system becomes unstable is controlled by the magnitude of the imaginary parts of these eigenvalues.” The wording here is strange. I would say that the threshold is of instability may be correlated with the imaginary parts of the eigenvalues, but not ‘controlled’.

Line 453. “Although our model incorporates a more realistic picture of microbial metabolism than previous consumer-resource theory…” Of any previous consumer-resource theory? Is that true?

Line 462-464. “However, allowing non-limiting resources to contribute to consumer biomass greatly complicates our mathematical analysis”. This statement sort of contradicts the ‘more realistic picture’ one above.

Line 465. “we have also implicitly assumed that the consumer can deplete many of the available resources.” From Eq. (1) is seems they consume all available resources, as there is no washout of resources.

Line 568-571. “If instead B has eigenvalues with non-zero imaginary part, then the clouds in the spectrum of J will have non-zero width, and some of these eigenvalues will cross the imaginary axis at a small value of n, creating an unstable fixed point.” That is really interesting, but why would B having non-zero imaginary part cause some of the values to cross the imaginary axis? There has to be an underlying reason that can be explained.

I would like to be positive, as the authors are certainly doing ambitious work. But this is a long and (for me) difficult paper; difficult, that is, to keep up with all the assumptions that the authors make, some of which do not seem reasonable, or are at least not justified here. My perseverance was running out about midway through the Discussion of my second reading of the manuscript. There is a lot or repetition. This manuscript reads like an unedited first draft and needs to be streamlined and ways found to get through the complexity. I am not a specialist in randomly assembled webs, but I have considerable experience in reading in this area, and I think this needs to be revised in a very major way; in fact, it needs to be rethought. Also, I don’t know what the criteria are for PLOS Computational Biology papers are, but I don’t see what is new computationally here. As far as I can see, computations are not being used in a novel way. Perhaps the authors could point out what is qualitatively new.

**Have the authors made all data and (if applicable) computational code underlying the findings in their manuscript fully available?**

Reviewer #1: Yes

Reviewer #2: Yes

Reviewer #3: None

PLOS authors have the option to publish the peer review history of their article (what does this mean?). If published, this will include your full peer review and any attached files.

Reviewer #1: No

Reviewer #2: No

Reviewer #3: No
---

## [Decision Letter · Decision Letter 1]

1 Jul 2022

Dear Mr. Gibbs,

Thank you very much for submitting your manuscript "Stability criteria for the consumption and exchange of essential resources" for consideration at PLOS Computational Biology. As with all papers reviewed by the journal, your manuscript was reviewed by members of the editorial board and by several independent reviewers. The reviewers appreciated the attention to an important topic. Based on the reviews, we are likely to accept this manuscript for publication, providing that you modify the manuscript according to the review recommendations.

Sincerely,

Jacopo Grilli

Associate Editor

PLOS Computational Biology

Natalia Komarova

Deputy Editor

PLOS Computational Biology

[LINK]

Reviewer's Responses to Questions

**Comments to the Authors:**

Reviewer #1: I appreciate the authors' efforts in answering my questions. I do not think my main concerns are resolved.

In my perspective, biological significance means the predictions made by the model are closely related to experimentally observed patterns. I do not think the stability criterion in this paper belongs to this category. It depends on all interaction matrix elements, and it is infeasible to measure them in the experiment.

Second, I agree that the stability criterion may be generalized to the case that the consumer abundances are not identical. But it still requires all resource abundances to be equal, and each species must consume one unique resource(otherwise, the resource abundance is unbound), making the model far from real ecosystems.

The instability at low consumer abundances is very interesting, and the results are of interest to theoretical ecologists. Unfortunately, I still think it fits more into a theoretical journal than PLOS computational biology.

Another comment:

From my understanding, the stability criterion is a sufficient condition for instability. If it is violated, it is wrong. So it is inappropriate to just use the average value in Fig 3, which clearly shows that it fails for the unstructured case in Appendix Fig C. To avoid confusion, I strongly suggest that the authors add error bars and Figure C in the appendix to Fig 3.

Reviewer #2: Reviewer #2 from the original submission here. The authors have addressed all my comments well, so I am in favor of this manuscript being accepted with no further revision needed. It will make a nice contribution to the burgeoning field of microbial cross-feeding and community stability.

Reviewer #3: The authors model an arbitrarily large number of microbial species feeding on an equal number of resources. The chemostat model includes cross-feeding, or the use by one organism of metabolites released by other organisms. The authors find conditions for feasibility and stability of the system that depend on structural aspects of the system.

Comments

I reviewed an earlier version of this manuscript. The revised version is much more clearly written and some changes have been made in the model. In particular, all resources contribute to biomass of a consumer, which is more realistic than the original assumption. The changes in the model are explained well in a response to Comment 1 of Reviewer 2, and in the revised text (e.g., Figure 1). This aspect of the model now seems more reasonable. It is also important that it is now stated that the equilibrium is unique and that both criteria for stability (Lines 201-202) are needed for ‘stability to be guaranteed’ (although it is not clear to me whether that means these are necessary and not just sufficient conditions). It is also important that that stability conditions are shown to hold for interaction networks that are only approximately symmetric. Evolution toward symmetry, that is, toward a more stable system also seems plausible. The authors have also added more support for their assumptions. Overall, this is a greatly improved manuscript.

I have some additional comments, most of which are minor.

Lines 36-49. The paper of Elser et al. (1996) seems relevant and could be cited here or somewhere: Elser, J.J., Dobberfuhl, D.R., MacKay, N.A. and Schampel, J.H., 1996. Organism size, life history, and N: P stoichiometry: toward a unified view of cellular and ecosystem processes. BioScience, 46(9), pp.674-684.

Line 59. Change ‘which which’ to ‘which’

Line 91-93. Is this statement correct:. “Under Liebig’s law, each consumer’s growth rate is determined only by the resource which contributes least to the production of new biomass.”? I think it is better stated that, under Liebig’s law, the consumer’s growth rate is determined by the resource that is available in the environment in the least amount relative to what is needed for the consumer’s growth.

Lines 111-112. “the excess intake of these resources is cycled back to the environment, possibly following some internal reactions that transform it.” Also, in Lines 118-120. “Before being secreted, this internal excess may be converted to other forms through intracellular reactions that do not create new biomass.” It is not clear to me whether a single resource is transformed by the ‘internal reactions’ to a new resource, or whether some combination of the resources that were taken up are combined chemically to a new resource out of more basic elements. The latter sounds much more complicated than what is going on in the equations, but I am not sure. Total biomass is being conserved, but it is not clear that basic building blocks (bio-elements) are being conserved. Lines 437-442 note that this aspect of the modeling will get more attention in the future.

Line 125. It sounds better to change ‘a constant fraction’ to ‘constant fractions’

Line 138. Change ‘negative real part’ to ‘negative real parts’

Line 201. Perhaps more could be said concerning matrix B. I realize that it is discussed in the Supplementary Information (2. Stability Criteria), but maybe more information can be given here. Matrix B comes from Equation (3) when resource diagonal matrix R has equal values r for all elements and matrix C_d is a diagonal matrix. A negative sign is also applied. If symmetric and negative, then all of the eigenvalues of B should be real and negative – or at least non-positive, I believe. So is condition 2 (Line 202) necessary, as the authors state in responses to review comments? I may be missing something.

Line 242. It is perhaps not necessary to say “In the main text”

Lines 309-310. “low consumer abundances lead to instability”. Low consumer abundances result from small values of input, ro, which usually relates to the high residence time of resources in the system. It is known, at least for simple model systems, that the shorter the residence time is, the more resilient the system is; e.g., DeAngelis, D.L., 1980. Energy flow, nutrient cycling, and ecosystem resilience. Ecology, 61(4), pp.764-771.. The effect of high throughflow (low residence time) could make unstructured systems here less vulnerable to the occurrence of local instability by pushing real parts of eigenvalues to larger negative values.

Lines 369-376. It is shown that the imaginary parts of the eigenvalues of B to the minimum resource inflow level at which the community can exist. Is this simply intuitive; i.e., as fluctuations would more seriously affect stability (possibly cause population values to become negative) if average population abundances are small due to low resource inflow rates?

Line 375. Change ‘imaginary part’ to ‘imaginary parts’

Line 481. Change ‘they require’ to ‘it requires’

This is a vastly improved manuscript. The methods and results are clear and the limitations of the modeling are discussed. The authors have cited appropriate references and put their work into context. The four main results seem to be sound and interesting, especially that symmetric interaction structures are protected from instability at low resource flow. My final comment is perhaps the authors should emphasize more why this manuscript is aimed at PLOS Computational Biology. Certainly, computations are important in the simulations, but it is not clear why the manuscript is more appropriate for PLOS Computational Biology than say, any number of theoretical biology journals.

**Have the authors made all data and (if applicable) computational code underlying the findings in their manuscript fully available?**

Reviewer #1: Yes

Reviewer #2: None

Reviewer #3: Yes

PLOS authors have the option to publish the peer review history of their article (what does this mean?). If published, this will include your full peer review and any attached files.

Reviewer #1: No

Reviewer #2: No

Reviewer #3: No

Figure Files:

Data Requirements:

Reproducibility:

References:

---

## [Decision Letter · Decision Letter 2]

29 Aug 2022

Dear Mr. Gibbs,

We are pleased to inform you that your manuscript 'Stability criteria for the consumption and exchange of essential resources' has been provisionally accepted for publication in PLOS Computational Biology.

Best regards,

Jacopo Grilli

Academic Editor

PLOS Computational Biology

Natalia Komarova

Section Editor

PLOS Computational Biology

Reviewer's Responses to Questions

**Comments to the Authors:**

Reviewer #3: I have read the revised manuscript. The authors have responded satisfactorily to my comments.

**Have the authors made all data and (if applicable) computational code underlying the findings in their manuscript fully available?**

Reviewer #3: Yes

PLOS authors have the option to publish the peer review history of their article (what does this mean?). If published, this will include your full peer review and any attached files.

Reviewer #3: **Yes: **Donald L. DeAngelis

---

## [Editor Report · Acceptance letter]

5 Sep 2022

PCOMPBIOL-D-21-02201R2 

Stability criteria for the consumption and exchange of essential resources

Dear Dr Gibbs,

I am pleased to inform you that your manuscript has been formally accepted for publication in PLOS Computational Biology. Your manuscript is now with our production department and you will be notified of the publication date in due course.

With kind regards,

Zsofia Freund
